# Are GANs Created Equal? A Large-Scale Study

**Mario Lucic**[*]    **Karol Kurach**[*]    **Marcin Michalski**    **Olivier Bousquet**    **Sylvain Gelly**
Google Brain

## Abstract

Generative adversarial networks (GAN) are a powerful subclass of generative models. Despite a very rich research activity leading to numerous interesting GAN algorithms, it is still very hard to assess which algorithm(s) perform better than others. We conduct a neutral, multi-faceted large-scale empirical study on state-of-the art models and evaluation measures. We find that most models can reach similar scores with enough hyperparameter optimization and random restarts. This suggests that improvements can arise from a higher computational budget and tuning more than fundamental algorithmic changes. To overcome some limitations of the current metrics, we also propose several data sets on which precision and recall can be computed. Our experimental results suggest that future GAN research should be based on more systematic and objective evaluation procedures. Finally, we did not find evidence that any of the tested algorithms consistently outperforms the non-saturating GAN introduced in [9].

## 1   Introduction

Generative adversarial networks (GAN) are a powerful subclass of generative models and were successfully applied to image generation and editing, semi-supervised learning, and domain adaptation [22, 27]. In the GAN framework the model learns a deterministic transformation $G$ of a simple distribution $p_z$, with the goal of matching the data distribution $p_d$. This learning problem may be viewed as a two-player game between the *generator*, which learns how to generate samples which resemble real data, and a *discriminator*, which learns how to discriminate between real and *fake* data. Both players aim to minimize their own cost and the solution to the game is the Nash equilibrium where neither player can improve their cost unilaterally [9].

Various flavors of GANs have been recently proposed, both purely unsupervised [9, 1, 10, 5] as well as conditional [20, 21]. While these models achieve compelling results in specific domains, there is still no clear consensus on which GAN algorithm(s) perform objectively better than others. This is partially due to the lack of robust and consistent metric, as well as limited comparisons which put all algorithms on equal footage, including the computational budget to search over all hyperparameters. Why is it important? Firstly, to help the practitioner choose a better algorithm from a very large set. Secondly, to make progress towards better algorithms and their understanding, it is useful to clearly assess which modifications are critical, and which ones are only good on paper, but do not make a significant difference in practice.

The main issue with evaluation stems from the fact that one cannot explicitly compute the probability $p_g(x)$. As a result, classic measures, such as log-likelihood on the test set, cannot be evaluated. Consequently, many researchers focused on qualitative comparison, such as comparing the visual quality of samples. Unfortunately, such approaches are subjective and possibly misleading [8]. As a remedy, two evaluation metrics were proposed to quantitatively assess the performance of GANs. Both assume access to a pre-trained classifier. *Inception Score (IS)* [24] is based on the fact that a good model should generate samples for which, when evaluated by the classifier, the class distribution has low entropy. At the same time, it should produce diverse samples covering all classes. In contrast,

---

[*]Indicates equal authorship. Correspondence to {`lucic,kkurach`}@google.com.

*Fréchet Inception Distance* is computed by considering the difference in embedding of true and fake data [11]. Assuming that the coding layer follows a multivariate Gaussian distribution, the distance between the distributions is reduced to the Fréchet distance between the corresponding Gaussians.

**Our main contributions:** (1) We provide a fair and comprehensive comparison of the state-of-the-art GANs, and empirically demonstrate that nearly all of them can reach similar values of FID, given a high enough computational budget. (2) We provide strong empirical evidence[2] that to compare GANs it is necessary to report a summary of distribution of results, rather than the best result achieved, due to the randomness of the optimization process and model instability. (3) We assess the robustness of FID to mode dropping, use of a different encoding network, and provide estimates of the best FID achievable on classic data sets. (4) We introduce a series of tasks of increasing difficulty for which undisputed measures, such as precision and recall, can be approximately computed. (5) We open-sourced our experimental setup and model implementations at goo.gl/G8kf5J.

## 2 Background and Related Work

There are several ongoing challenges in the study of GANs, including their convergence and generalization properties [2, 19], and optimization stability [24, 1]. Arguably, the most critical challenge is their *quantitative* evaluation. The classic approach towards evaluating generative models is based on model likelihood which is often intractable. While the log-likelihood can be approximated for distributions on low-dimensional vectors, in the context of complex high-dimensional data the task becomes extremely challenging. Wu et al. [26] suggest an annealed importance sampling algorithm to estimate the hold-out log-likelihood. The key drawback of the proposed approach is the assumption of the Gaussian observation model which carries over all issues of kernel density estimation in high-dimensional spaces. Theis et al. [25] provide an analysis of common failure modes and demonstrate that it is possible to achieve high likelihood, but low visual quality, and vice-versa. Furthermore, they argue against using Parzen window density estimates as the likelihood estimate is often incorrect. In addition, ranking models based on these estimates is discouraged [4]. For a discussion on other drawbacks of likelihood-based training and evaluation consult Huszár [12].

**Inception Score (IS)**. Proposed by Salimans et al. [24], IS offers a way to quantitatively evaluate the quality of generated samples. The score was motivated by the following considerations: (i) The conditional label distribution of samples containing meaningful objects should have low entropy, and (ii) The variability of the samples should be high, or equivalently, the marginal $\int_z p(y|x = G(z))dz$ should have high entropy. Finally, these desiderata are combined into one score, $\texttt{IS}(G) = \exp(\mathbb{E}_{x \sim G}[d_{KL}(p(y \mid x), p(y)])$. The classifier is Inception Net trained on Image Net. The authors found that this score is well-correlated with scores from human annotators [24]. Drawbacks include insensitivity to the prior distribution over labels and not being a proper *distance*.

**Fréchet Inception Distance (FID).** Proposed by Heusel et al. [11], FID provides an alternative approach. To quantify the quality of generated samples, they are first embedded into a feature space given by (a specific layer) of Inception Net. Then, viewing the embedding layer as a continuous multivariate Gaussian, the mean and covariance is estimated for both the generated data and the real data. The Fréchet distance between these two Gaussians is then used to quantify the quality of the samples, i.e. $\texttt{FID}(x, g) = ||\mu_x - \mu_g||_2^2 + \text{Tr}(\Sigma_x + \Sigma_g - 2(\Sigma_x \Sigma_g)^{\frac{1}{2}})$, where $(\mu_x, \Sigma_x)$, and $(\mu_g, \Sigma_g)$ are the mean and covariance of the sample embeddings from the data distribution and model distribution, respectfully. The authors show that the score is consistent with human judgment and more robust to noise than IS [11]. Furthermore, the authors present compelling results showing negative correlation between the FID and visual quality of generated samples. Unlike IS, FID can detect intra-class mode dropping, i.e. a model that generates only one image per class can score a perfect IS, but will have a bad FID. We provide a thorough empirical analysis of FID in Section 5. A significant drawback of both measures is the inability to detect overfitting. A "memory GAN" which stores all training samples would score perfectly. Finally, as the FID estimator is *consistent*, relative model comparisons for large sample sizes are sound.

A very recent study comparing several GANs using IS has been presented by Fedus et al. [7]. The authors focus on IS and consider a smaller subset of GANs. In contrast, our focus is on providing a *fair assessment* of the current state-of-the-art GANs using FID, as well as precision and recall, and also verifying the robustness of these models in a large-scale empirical evaluation.

Table 1: Generator and discriminator loss functions. The main difference whether the discriminator outputs a probability (MM GAN, NS GAN, DRAGAN) or its output is unbounded (WGAN, WGAN GP, LS GAN, BEGAN), whether the gradient penalty is present (WGAN GP, DRAGAN) and where is it evaluated.

| GAN | DISCRIMINATOR LOSS | GENERATOR LOSS |
|---|---|---|
| MM GAN | $\mathcal{L}_{\mathrm{D}}^{\mathrm{GAN}} = -\mathbb{E}_{x\sim p_d}[\log(D(x))] - \mathbb{E}_{\hat{x}\sim p_g}[\log(1-D(\hat{x}))]$ | $\mathcal{L}_{\mathrm{G}}^{\mathrm{GAN}} = \mathbb{E}_{\hat{x}\sim p_g}[\log(1-D(\hat{x}))]$ |
| NS GAN | $\mathcal{L}_{\mathrm{D}}^{\mathrm{NSGAN}} = -\mathbb{E}_{x\sim p_d}[\log(D(x))] - \mathbb{E}_{\hat{x}\sim p_g}[\log(1-D(\hat{x}))]$ | $\mathcal{L}_{\mathrm{G}}^{\mathrm{NSGAN}} = -\mathbb{E}_{\hat{x}\sim p_g}[\log(D(\hat{x}))]$ |
| WGAN | $\mathcal{L}_{\mathrm{D}}^{\mathrm{WGAN}} = -\mathbb{E}_{x\sim p_d}[D(x)] + \mathbb{E}_{\hat{x}\sim p_g}[D(\hat{x})]$ | $\mathcal{L}_{\mathrm{G}}^{\mathrm{WGAN}} = -\mathbb{E}_{\hat{x}\sim p_g}[D(\hat{x})]$ |
| WGAN GP | $\mathcal{L}_{\mathrm{D}}^{\mathrm{WGANGP}} = \mathcal{L}_{\mathrm{D}}^{\mathrm{WGAN}} + \lambda\mathbb{E}_{\hat{x}\sim p_g}[(||\nabla D(\alpha x + (1-\alpha\hat{x}))||_2 - 1)^2]$ | $\mathcal{L}_{\mathrm{G}}^{\mathrm{WGANGP}} = -\mathbb{E}_{\hat{x}\sim p_g}[D(\hat{x})]$ |
| LS GAN | $\mathcal{L}_{\mathrm{D}}^{\mathrm{LSGAN}} = -\mathbb{E}_{x\sim p_d}[(D(x)-1)^2] + \mathbb{E}_{\hat{x}\sim p_g}[D(\hat{x})^2]$ | $\mathcal{L}_{\mathrm{G}}^{\mathrm{LSGAN}} = -\mathbb{E}_{\hat{x}\sim p_g}[(D(\hat{x}-1))^2]$ |
| DRAGAN | $\mathcal{L}_{\mathrm{D}}^{\mathrm{DRAGAN}} = \mathcal{L}_{\mathrm{D}}^{\mathrm{GAN}} + \lambda\mathbb{E}_{\hat{x}\sim p_d+\mathcal{N}(0,c)}[(||\nabla D(\hat{x})||_2 - 1)^2]$ | $\mathcal{L}_{\mathrm{G}}^{\mathrm{DRAGAN}} = \mathbb{E}_{\hat{x}\sim p_g}[\log(1-D(\hat{x}))]$ |
| BEGAN | $\mathcal{L}_{\mathrm{D}}^{\mathrm{BEGAN}} = \mathbb{E}_{x\sim p_d}[||x - \mathrm{AE}(x)||_1] - k_t\mathbb{E}_{\hat{x}\sim p_g}[||\hat{x} - \mathrm{AE}(\hat{x})||_1]$ | $\mathcal{L}_{\mathrm{G}}^{\mathrm{BEGAN}} = \mathbb{E}_{\hat{x}\sim p_g}[||\hat{x} - \mathrm{AE}(\hat{x})||_1]$ |

# 3 Flavors of Generative Adversarial Networks

In this work we focus on *unconditional* generative adversarial networks. In this setting, only unlabeled data is available for learning. The optimization problems arising from existing approaches differ by (i) the constraint on the discriminators output and corresponding loss, and the presence and application of gradient norm penalty.

In the original GAN formulation [9] two loss functions were proposed. In the *minimax* GAN the discriminator outputs a probability and the loss function is the negative log-likelihood of a binary classification task (MM GAN in Table 1). Here the generator learns to generate samples that have a low probability of being fake. To improve the gradient signal, the authors also propose the *non-saturating* loss (NS GAN in Table 1), where the generator instead aims to maximize the probability of generated samples being real. In Wasserstein GAN [1] the discriminator is allowed to output a real number and the objective function is equivalent to the MM GAN loss without the sigmoid (WGAN in Table 1). The authors prove that, under an optimal (Lipschitz smooth) discriminator, minimizing the value function with respect to the generator minimizes the Wasserstein distance between model and data distributions. Weights of the discriminator are clipped to a small absolute value to enforce smoothness. To improve on the stability of the training, Gulrajani et al. [10] instead add a soft constraint on the norm of the gradient which encourages the discriminator to be 1-Lipschitz. The gradient norm is evaluated on points obtained by linear interpolation between data points and generated samples where the optimal discriminator should have unit gradient norm [10]. Gradient norm penalty can also be added to both MM GAN and NS GAN and evaluated around the data manifold (DRAGAN [15] in Table 1 based on NS GAN). This encourages the discriminator to be piecewise linear around the data manifold. Note that the gradient norm can also be evaluated between fake and real points, similarly to WGAN GP, and added to either MM GAN or NS GAN [7]. Mao et al. [18] propose a least-squares loss for the discriminator and show that minimizing the corresponding objective (LS GAN in Table 1) implicitly minimizes the Pearson $\chi^2$ divergence. The idea is to provide smooth loss which saturates slower than the sigmoid cross-entropy loss of the original MM GAN. Finally, Berthelot et al. [5] propose to use an autoencoder as a discriminator and optimize a lower bound of the Wasserstein distance between auto-encoder *loss distributions* on real and fake data. They introduce an additional hyperparameter $\gamma$ to control the equilibrium between the generator and discriminator.

# 4 Challenges of a Fair Comparison

There are several interesting dimensions to this problem, and there is no single *right way* to compare these models (i.e. the loss function used in each GAN). Unfortunately, due to the combinatorial explosion in the number of choices and their ordering, not all relevant options can be explored. While there is *no definite answer* on how to best compare two models, in this work we have made several pragmatic choices which were motivated by two practical concerns: providing a neutral and fair comparison, and a hard limit on the computational budget.

**Which metric to use?** Comparing models implies access to some metric. As discussed in Section 2, classic measures, such as model likelihood cannot be applied. We will argue for and study two sets of

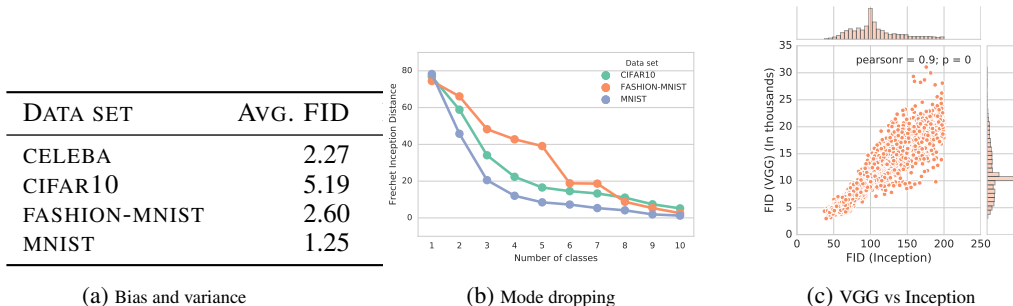

| DATA SET | AVG. FID |
|---|---|
| CELEBA | 2.27 |
| CIFAR10 | 5.19 |
| FASHION-MNIST | 2.60 |
| MNIST | 1.25 |

(a) Bias and variance  (b) Mode dropping  (c) VGG vs Inception

Figure 1: Figure (a) shows that FID has a slight bias, but low variance on samples of size 10000. Figure (b) shows that FID is extremely sensitive to mode dropping. Figure (c) shows the high rank correlation (Spearman's $\rho = 0.9$) between FID score computed on InceptionNet vs FID computed using VGG for the CELEBA data set (for interesting range: FID $< 200$).

evaluation metrics in Section 5: FID, which can be computed on all data sets, and precision, recall, and $F_1$, which we can compute for the proposed tasks.

**How to compare models?** Even when the metric is fixed, a given algorithm can achieve very different scores, when varying the architecture, hyperparameters, random initialization (i.e. random seed for initial network weights), or the data set. Sensible targets include best score across all dimensions (e.g. to claim the best performance on a fixed data set), average or median score (rewarding models which are good in expectation), or even the worst score (rewarding models with worst-case robustness). These choices can even be combined — for example, one might train the model multiple times using the best hyperparameters, and average the score over random initializations).

For each of these dimensions, we took several pragmatic choices to reduce the number of possible configurations, while still exploring the most relevant options.

1. **Architecture**: We use the *same* architecture for all models. We note that this architecture suffices to achieve good performance on considered data sets.
2. **Hyperparameters**: For both training hyperparameters (e.g. the learning rate), as well as model specific ones (e.g. gradient penalty multiplier), there are two valid approaches: (i) perform the hyperparameter optimization for each data set, or (ii) perform the hyperparameter optimization on one data set and *infer* a good range of hyperparameters to use on other data sets. We explore both avenues in Section 6.
3. **Random seed**: Even with everything else being fixed, varying the random seed may influence on the results. We study this effect and report the corresponding confidence intervals.
4. **Data set**: We chose four popular data sets from GAN literature.
5. **Computational budget**: Depending on the budget to optimize the parameters, different algorithms can achieve the best results. We explore how the results vary depending on the budget $k$, where $k$ is the number of hyperparameter settings for a fixed model.

In practice, one can either use hyperparameter values suggested by respective authors, or try to optimize them. Figure 4 and in particular Figure 14 show that optimization is necessary. Hence, we optimize the hyperparameters for each model and data set by performing a random search. While we present the results which were obtained by a random search, we have also investigated sequential Bayesian optimization, which resulted in comparable results. We concur that the models with fewer hyperparameters have an advantage over models with many hyperparameters, but consider this fair as it reflects the experience of practitioners searching for good hyperparameters for their setting.

## 5 Metrics

In this work we focus on two sets of metrics. We first analyze the recently proposed FID in terms of robustness (of the metric itself), and conclude that it has desirable properties and can be used in practice. Nevertheless, this metric, as well as Inception Score, is incapable of detecting overfitting: a *memory GAN* which simply stores all training samples would score perfectly under both measures. Based on these shortcomings, we propose an approximation to precision and recall for GANs and how that it can be used to quantify the degree of overfitting. We stress that the proposed method should be viewed as complementary to IS or FID, rather than a replacement.

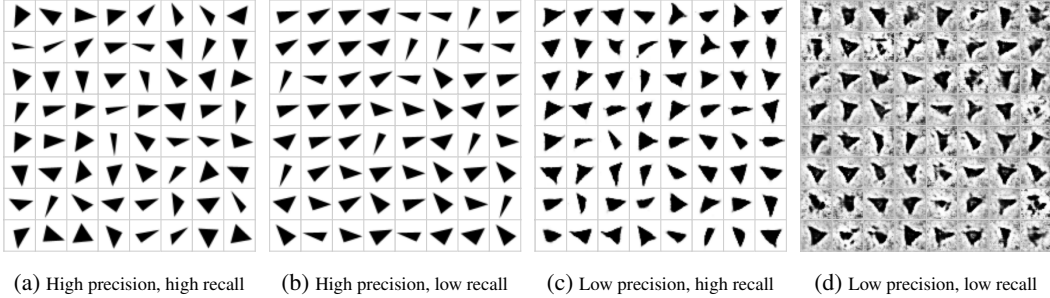

(a) High precision, high recall    (b) High precision, low recall    (c) Low precision, high recall    (d) Low precision, low recall

Figure 2: Samples from models with (a) high recall and precision, (b) high precision, but low recall (lacking in diversity), (c) low precision, but high recall (can decently reproduce triangles, but fails to capture convexity), and (d) low precision and low recall.

**Fréchet Inception Distance.** FID was shown to be robust to noise [11]. Here we quantify the bias and variance of FID, its sensitivity to the encoding network and sensitivity to mode dropping. To this end, we partition the data set into two groups, i.e. $\mathcal{X} = \mathcal{X}_1 \cup \mathcal{X}_2$. Then, we define the data distribution $p_d$ as the empirical distribution on a random subsample of $\mathcal{X}_1$ and the model distribution $p_g$ to be the empirical distribution on a random subsample from $\mathcal{X}_2$. For a random partition this "model distribution" should follow the data distribution.

We evaluate the bias and variance of FID on four data sets from the GAN literature. We start by using the default train vs. test partition and compute the FID between the test set (limited to $N = 10000$ samples for CelebA) and a sample of size $N$ from the train set. Sampling from the train set is performed $M = 50$ times. The optimistic estimates of FID are reported in Table 1. We observe that FID has high bias, but small variance. From this perspective, estimating the full covariance matrix might be unnecessary and counter-productive, and a constrained version might suffice. To test the sensitivity to train vs. test partitioning, we consider 50 random partitions (keeping the relative sizes fixed, i.e. $6 : 1$ for MNIST) and compute the FID with $M = 1$ sample. We observe results similar to Table 1 which is expected as both training and testing data sets are sampled from the same distribution. Furthermore, we evaluate the sensitivity to mode dropping as follows: we fix a partition $\mathcal{X} = \mathcal{X}_1 \cup \mathcal{X}_2$ and subsample $\mathcal{X}_2$ while keeping only samples from the first $k$ classes, increasing $k$ from 1 to 10. For each $k$, we consider 50 random subsamples from $\mathcal{X}_2$. Figure 1 shows that FID is heavily influenced by the missing modes. Finally, we estimate the sensitivity to the choice of the encoding network by computing FID using the 4096 dimensional FC7 layer of the VGG network trained on ImageNet. Figure 1 shows the resulting distribution. We observe high Spearman's rank correlation ($\rho = 0.9$) which encourages the use of the coding layer suggested by the authors.

**Precision, recall and $F_1$ score.** Precision, recall and $F_1$ score are proven and widely adopted techniques for quantitatively evaluating the quality of discriminative models. Precision measures the fraction of relevant retrieved instances among the retrieved instances, while recall measures the fraction of the retrieved instances among relevant instances. $F_1$ score is the harmonic average of precision and recall. Notice that IS mainly captures precision: It will not penalize the model for not producing all modes of the data distribution — it will only penalize the model for not producing all *classes*. On the other hand, FID captures both precision and recall. Indeed, a model which fails to recover different modes of the data distribution will suffer in terms of FID.

We propose a simple and effective data set for evaluating (and comparing) generative models. Our main motivation is that the currently used data sets are either too simple (e.g. simple mixtures of Gaussians, or MNIST) or too complex (e.g. ImageNet). We argue that it is critical to be able to increase the complexity of the task in a relatively smooth and controlled fashion. To this end, we present a set of tasks for which we can *approximate* the precision and recall of each model. As a result, we can compare different models based on established metrics. The main idea is to construct a data manifold such that the distances from samples to the manifold can be computed efficiently. As a result, the problem of evaluating the quality of the generative model is effectively transformed into a problem of computing the distance to the manifold. This enables an intuitive approach for defining the quality of the model. Namely, if the samples from the model distribution $p_g$ are (on average) close to the manifold, its *precision* is high. Similarly, high *recall* implies that the generator can recover (i.e. generate something close to) any sample from the manifold.

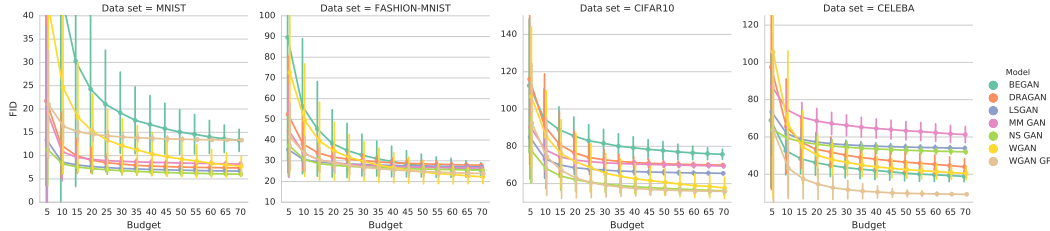

Figure 3: How does the minimum FID behave as a function of the budget? The plot shows the distribution of the minimum FID achievable for a fixed budget along with one standard deviation interval. For each budget, we estimate the mean and variance using 5000 bootstrap resamples out of 100 runs. We observe that, given a relatively low budget, all models achieve a similar minimum FID. Furthermore, for a fixed FID, "bad" models can outperform "good" models given enough computational budget. We argue that the computational budget to search over hyperparameters is an important aspect of the comparison between algorithms.

For general data sets, this reduction is impractical as one has to compute the distance to the manifold which we are trying to learn. However, if we *construct a manifold* such that this distance is efficiently computable, the precision and recall can be efficiently evaluated. To this end, we propose a set of toy data sets for which such computation can be performed efficiently: The manifold of convex polygons. As the simplest example, let us focus on gray-scale triangles represented as one channel images as in Figure 2. These triangles belong to a low-dimensional manifold $\mathcal{C}_3$ embedded in $\mathbb{R}^{d \times d}$. Intuitively, the coordinate system of this manifold represents the axes of variation (e.g. rotation, translation, minimum angle size, etc.). A good generative model should be able to capture these factors of variation and recover the training samples. Furthermore, it should recover any sample from this manifold from which we can efficiently sample which is illustrated in Figure 2.

**Computing the distance to the manifold.** Let us consider the simplest case: single-channel gray scale images represented as vectors $x \in \mathbb{R}^{d^2}$. The distance of a sample $\hat{x} \in \mathbb{R}^{d^2}$ to the manifold is defined as the squared Euclidean distance to the closest sample from the manifold $\mathcal{C}_3$, i.e. $\min_{x \in \mathcal{C}_3} \ell(x, \hat{x}) = \sum_{i=1}^{d^2} ||x_i - \hat{x}_i||_2^2$. This is a non-convex optimization problem. We find an approximate solution by gradient descent on the vertices of the triangle (more generally, a convex polygon), ensuring that each iterate is a valid triangle (more generally, a convex polygon). To reduce the false-negative rate we repeat the algorithm 5 times from random initial solutions. To compute the latent representation of a sample $\hat{x} \in \mathbb{R}^{d \times d}$ we *invert* the generator, i.e. we solve $z^\star = \arg\min_{z \in \mathbb{R}^{d_z}} ||\hat{x} - G(z)||_2^2$, using gradient descent on $z$ while keeping G fixed [17].

## 6 Large-scale Experimental Evaluation

We consider two budget-constrained experimental setups whereby in the (i) **wide one-shot setup** one may select 100 samples of hyper-parameters per model, and where the range for each hyperparameter is *wide*, and (ii) the **narrow two-shots setup** where one is allowed to select 50 samples from more narrow ranges which were manually selected by first performing the wide hyperparameter search over a specific data set. For the exact ranges and hyperparameter search details we refer the reader to the Appendix A. In the second set of experiments we evaluate the models based on the "novel" metric: $F_1$ score on the proposed data set. Finally, we included the Variational Autoencoder [14] in the experiments as a popular alternative.

**Experimental setup.** To ensure a fair comparison, we made the following choices: (i) we use the generator and discriminator architecture from INFO GAN [6] as the resulting function space is rich enough and all considered GANs were not originally designed for this architecture. Furthermore, it is similar to a proven architecture used in DCGAN [22]. The exception is BEGAN where an autoencoder is used as the discriminator. We maintain similar expressive power to INFO GAN by using identical convolutional layers the encoder and approximately matching the total number of parameters.

For all experiments we fix the latent code size to 64 and the prior distribution over the latent space to be uniform on $[-1, 1]^{64}$, except for VAE where it is Gaussian $\mathcal{N}(0, \mathbf{I})$. We choose Adam [13] as the optimization algorithm as it was the most popular choice in the GAN literature (cf. Appendix F for an empirical comparison to RMSProp). We apply the same learning rate for both generator and discriminator. We set the batch size to 64 and perform optimization for 20 epochs on MNIST

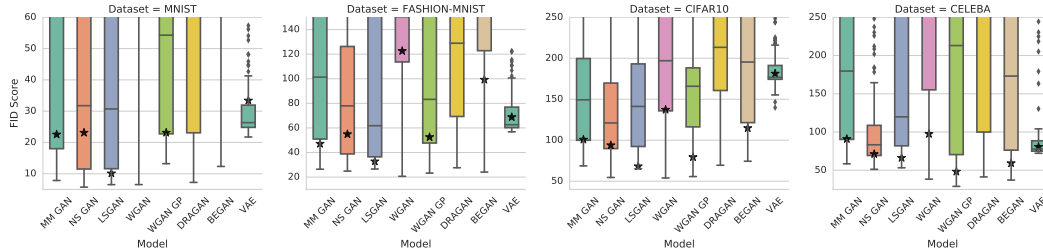

Figure 4: A *wide range* hyperparameter search (100 hyperparameter samples per model). Black stars indicate the performance of suggested hyperparameter settings. We observe that GAN training is extremely sensitive to hyperparameter settings and there is no model which is significantly more stable than others.

and FASHION MNIST, 40 on CELEBA and 100 on CIFAR. These data sets are a popular choice for generative modeling, range from simple to medium complexity, which makes it possible to run many experiments as well as getting decent results.

Finally, we allow for recent suggestions, such as batch normalization in the discriminator, and imbalanced update frequencies of generator and discriminator. We explore these possibilities, together with learning rate, parameter $\beta_1$ for ADAM, and hyperparameters of each model. We report the hyperparameter ranges and other details in Appendix A.

**A large hyperparameter search.** We perform hyperparameter optimization and, for each run, look for the *best* FID across the training run (simulating early stopping). To choose the *best* model, every 5 epochs we compute the FID between the 10k samples generated by the model and the 10k samples from the test set. We have performed this computationally expensive search for each data set. We present the sensitivity of models to the hyper-parameters in Figure 4 and the best FID achieved by each model in Table 2. We compute the best FID, in two phases: We first run a large-scale search on a wide range of hyper-parameters, and select the best model. Then, we re-run the training of the selected model 50 times with different initialization seeds, to estimate the stability of the training and report the mean FID and standard deviation, excluding outliers.

Furthermore, we consider the *mean* FID as the computational budget increases which is shown in Figure 3. There are three important observations. Firstly, there is no algorithm which clearly dominates others. Secondly, for an interesting range of FIDs, a "bad" model trained on a large budget can out perform a "good" model trained on a small budget. Finally, when the budget is limited, any statistically significant comparison of the models is unattainable.

**Impact of limited computational budget.** In some cases, the computational budget available to a practitioner is too small to perform such a large-scale hyperparameter search. Instead, one can tune the range of hyperparameters on one data set and interpolate the good hyperparameter ranges for other data sets. We now consider this setting in which we allow only 50 samples from a set of narrow ranges, which were selected based on the wide hyperparameter search on the FASHION-MNIST data set. We report the narrow hyperparameter ranges in Appendix A. Figure 14 shows the variance of FID per model, where the hyperparameters were selected from narrow ranges. From the practical point of view, there are significant differences between the models: in some cases the hyperparameter

Table 2: Best FID obtained in a large-scale hyperparameter search for each data set. The asterisk (*) on some combinations of models and data sets indicates the presence of significant outlier runs, usually severe mode collapses or training failures (** indicates up to 20% failures). We observe that the performance of each model heavily depends on the data set and no model strictly dominates the others. Note that these results are **not "state-of-the-art"**: (i) larger architectures could improve all models, (ii) authors often report the best FID which opens the door for random seed optimization.

|  | MNIST | FASHION | CIFAR | CELEBA |
|---|---|---|---|---|
| MM GAN | $9.8 \pm 0.9$ | $29.6 \pm 1.6$ | $72.7 \pm 3.6$ | $65.6 \pm 4.2$ |
| NS GAN | $6.8 \pm 0.5$ | $26.5 \pm 1.6$ | $58.5 \pm 1.9$ | $55.0 \pm 3.3$ |
| LSGAN | $7.8 \pm 0.6*$ | $30.7 \pm 2.2$ | $87.1 \pm 47.5$ | $53.9 \pm 2.8*$ |
| WGAN | $6.7 \pm 0.4$ | $21.5 \pm 1.6$ | $55.2 \pm 2.3$ | $41.3 \pm 2.0$ |
| WGAN GP | $20.3 \pm 5.0$ | $24.5 \pm 2.1$ | $55.8 \pm 0.9$ | $30.0 \pm 1.0$ |
| DRAGAN | $7.6 \pm 0.4$ | $27.7 \pm 1.2$ | $69.8 \pm 2.0$ | $42.3 \pm 3.0$ |
| BEGAN | $13.1 \pm 1.0$ | $22.9 \pm 0.9$ | $71.4 \pm 1.6$ | $38.9 \pm 0.9$ |
| VAE | $23.8 \pm 0.6$ | $58.7 \pm 1.2$ | $155.7 \pm 11.6$ | $85.7 \pm 3.8$ |

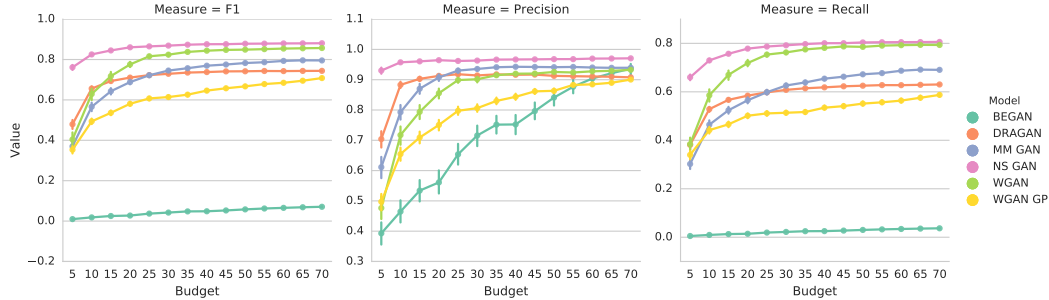

Figure 5: How does $F_1$ score vary with computational budget? The plot shows the distribution of the maximum $F_1$ score achievable for a fixed budget with a 95% confidence interval. For each budget, we estimate the mean and confidence interval (of the mean) using 5000 bootstrap resamples out of 100 runs. When optimizing for $F_1$ score, both NS GAN and WGAN enjoy high precision and recall. The underwhelming performance of BEGAN and VAE on this particular data set merits further investigation.

ranges *transfer* from one data set to the others (e.g. NS GAN), while others are more sensitive to this choice (e.g. WGAN). We note that better scores can be obtained by a wider hyperparameter search. These results supports the conclusion that discussing the *best* score obtained by a model on a data set is not a meaningful way to discern between these models. One should instead discuss the distribution of the obtained scores.

**Robustness to random initialization.** For a fixed model, hyperparameters, training algorithm, and the order that the data is presented to the model, one would expect similar model performance. To test this hypothesis we re-train the best models from the limited hyperparameter range considered for the previous section, while changing the initial weights of the generator and discriminator networks (i.e. by varying a random seed). Table 2 and Figure 15 show the results for each data set. Most models are relatively robust to random initialization, except LSGAN, even though for all of them the variance is significant and should be taken into account when comparing models.

**Precision, recall, and $F_1$.** We perform a search over the wide range of hyperparameters and compute precision and recall by considering $n = 1024$ samples. In particular, we compute the precision of the model by computing the fraction of generated samples with distance below a threshold $\delta = 0.75$. We then consider $n$ samples from the test set and invert each sample $x$ to compute $z^\star = G^{-1}(x)$ and compute the squared Euclidean distance between $x$ and $G(z^\star)$. We define the recall as the fraction of samples with squared Euclidean distance below $\delta$. Figure 5 shows the results where we select the best $F_1$ score for a fixed model and hyperparameters and vary the budget. We observe that even for this seemingly simple task, many models struggle to achieve a high $F_1$ score. Analogous plots where we instead maximize precision or recall for various thresholds are presented in Appendix E.

# 7 Limitations of the Study

**Data sets, neural architectures, and optimization issues.** While we consider classic data sets from GAN research, unconditional generation was recently applied to data sets of higher resolution and arguably higher complexity. In this study we use one neural network architecture which suffices to achieve good results in terms of FID on all considered data sets. However, given data sets of higher complexity and higher resolution, it might be necessary to significantly increase the number of parameters, which in turn might lead to larger quantitative differences between different methods. Furthermore, different objective functions might become sensible to the choice of the optimization method, the number of training steps, and possibly other optimization hyperparameters. These effects should be systematically studied in future work.

**Metrics.** It remains to be examined whether FID is stable under a more radical change of the encoding, e.g using a network trained on a different task. Furthermore, it might be possible to "fool" FID can probably by introducing artifacts specialized to the encoding network. From the classic machine learning point of view, a major drawback of FID is that it cannot detect overfitting to the training data set – an algorithm that outputs only the training examples would have an excellent score. As such, developing quantitative evaluation metrics is a critical research direction [3, 23].

**Exploring the space of hyperparameters.** Ideally, hyperparameter values suggested by the authors should transfer across data sets. As such, exploring the hyperparameters "close" to the suggested ones is a natural and valid approach. However, Figure 4 and in particular Figure 14 show that optimization is necessary. In addition, such an approach has several drawbacks: (a) no recommended hyperparameters are available for a given data set, (b) the parameters are different for each data set, (c) several popular models have been tuned by the community, which might imply an unfair comparison. Finally, instead of random search it might be beneficial to apply (carefully tuned) sequential Bayesian optimization which is computationally beyond the scope of this study, but nevertheless a great candidate for future work [16].

## 8    Conclusion

In this paper we have started a discussion on how to neutrally and fairly compare GANs. We focus on two sets of evaluation metrics: (i) The Fréchet Inception Distance, and (ii) precision, recall and $F_1$. We provide empirical evidence that FID is a reasonable metric due to its robustness with respect to mode dropping and encoding network choices. Our main insight is that to compare models it is meaningless to report the *minimum* FID achieved. Instead, we propose to compare distributions of the minimum achivable FID for a fixed computational budget. Indeed, empirical evidence presented herein imply that algorithmic differences in state-of-the-art GANs become less relevant, as the computational budget increases. Furthermore, given a limited budget (say a month of compute-time), a "good" algorithm might be outperformed by a "bad" algorithm.

As discussed in Section 4, many dimensions have to be taken into account for model comparison, and this work only explores a subset of the options. We cannot exclude the possibility that that some models significantly outperform others under currently unexplored conditions. Nevertheless, notwithstanding the limitations discussed in Section 7, this work strongly suggests that future GAN research should be more experimentally systematic and model comparison should be performed on neutral ground.

## Acknowledgments

We would like to acknowledge Tomas Angles for advocating convex polygons as a benchmark data set. We would like to thank Ian Goodfellow, Michaela Rosca, Ishaan Gulrajani, David Berthelot, and Xiaohua Zhai for useful discussions and remarks.

## Footnotes

[2]Reproducing these experiments requires approximately 6.85 GPU years (NVIDIA P100).

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
