[Supplementary Material]

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

# A  Wide and narrow hyperparameter ranges

The *wide* and *narrow* ranges of hyper-parameters are presented in Table 3 and Table 4 respectively. In both tables, U(a, b) means that the variable was sample uniformly from the range $[a, b]$. The L(a, b) means that that the variable was sampled on a log-scale, that is $x \, L(a, b) \iff x \, 10^{U(log(a), log(b))}$. The parameters used in the search:

- $\beta_1$: the parameter of the Adam optimization algorithm.
- Learning rate: generator/discriminator learning rate.
- $\lambda$: Multiplier of the gradient penalty for DRAGAN and WGAN GP. Learning rate for $k_t$ in BEGAN.
- Disc iters: Number of discriminator updates per one generator update.
- batchnorm: If True, the batch normalization will be used in the discriminator.
- $\gamma$: Parameter of BEGAN.
- clipping: Parameter of WGAN, weights will be clipped to this value.

Table 3: Wide ranges of hyper-parameters used for the large-scale search. "U" denotes uniform sampling, "L" sampling on a log-scale.

|  | MM GAN | NS GAN | LSGAN | WGAN | WGAN GP | DRAGAN | BEGAN | VAE |
|---|---|---|---|---|---|---|---|---|
| Adam's $\beta_1$ |  |  |  | U(0, 1) |  |  |  |  |
| learning rate |  |  |  | $L(10^{-5}, 10^{-2})$ |  |  |  |  |
| $\lambda$ | N/A | N/A | N/A | N/A | $L(10^{-1}, 10^2)$ | $L(10^{-1}, 10^2)$ | $L(10^{-4}, 10^{-2})$ | N/A |
| disc iter |  |  |  | Either 1 or 5, sampled with the same probablity |  |  |  |  |
| batchnorm |  |  |  | True or False, sampled with the same probability |  |  |  |  |
| $\gamma$ | N/A | N/A | N/A | N/A | N/A | N/A | U(0, 1) | N/A |
| clipping | N/A | N/A | N/A | $L(10^{-3}, 10^0)$ | N/A | N/A | N/A | N/A |

Table 4: Narrow ranges of hyper-parameters used in the search with 50 samples per model. The ranges were optimized by looking at the wide search results for fashion-mnist data set. "U" denotes uniform sampling, "L" sampling on a log-scale.

|  | MM GAN | NS GAN | LSGAN | WGAN | WGAN GP | DRAGAN | BEGAN | VAE |
|---|---|---|---|---|---|---|---|---|
| Adam's $\beta_1$ |  |  |  | Always 0.5 |  |  |  |  |
| learning rate |  |  |  | $L(10^{-4}, 10^{-3})$ |  |  |  |  |
| $\lambda$ | N/A | N/A | N/A | N/A | $L(10^{-1}, 10^1)$ | $L(10^{-1}, 10^1)$ | $L(10^{-4}, 10^{-2})$ | N/A |
| disc iter |  |  |  | Always 1 |  |  |  |  |
| batchnorm | True/False | True/False | True/False | True/False | False | False | True/False | True/False |
| $\gamma$ | N/A | N/A | N/A | N/A | N/A | N/A | U(0.6, 0.9) | N/A |
| clipping | N/A | N/A | N/A | $L(10^{-2}, 10^0)$ | N/A | N/A | N/A | N/A |

# B  Which parameters really matter?

Figure 6, Figure 7, Figure 8 and Figure 9 present scatter plots for data sets FASHION MNIST, MNIST, CIFAR, CELEBA respectively. For each model and hyper-parameter we estimate its impact on the final FID. Figure 6 was used to select narrow ranges of hyper-parameters.

# FASHION-MNIST (wide range)

Figure 6: Wide range scatter plots for FASHION MNIST. For each algorithm (column) and each parameter (row), the corresponding scatter plot shows the FID in function of the parameter. This illustrates the sensitivity of each algorithm w.r.t. each parameter. Those results have been used to choose the *narrow range* in Table 4. For example, Adam's $\beta_1$ does not seem to significantly impact any algorithm, so for the narrow range, we fix its value to always be $0.5$. Likewise, we fix the number of discriminator iterations to be always 1. For other parameters, the selected range is smaller (e.g. learning rate), or can differ for each algorithm (e.g. batch norm).

# MNIST (wide range)

Figure 7: Wide range scatter plots for MNIST. For each algorithm (column) and each parameter (row), the corresponding scatter plot shows the FID in function of the parameter. This illustrates the sensitivity of each algorithm w.r.t. each parameter.

# CIFAR10 (wide range)

Figure 8: Wide range scatter plots for CIFAR10. For each algorithm (column) and each parameter (row), the corresponding scatter plot shows the FID in function of the parameter. This illustrates the sensitivity of each algorithm w.r.t. each parameter.

# CELEBA (wide range)

Figure 9: Wide range scatter plots for CELEBA. For each algorithm (column) and each parameter (row), the corresponding scatter plot shows the FID in function of the parameter. This illustrates the sensitivity of each algorithm w.r.t. each parameter.

# C  Fréchet Inception Distance and Image Quality

It is interesting to see how the FID translates to the image quality. In Figure 10, Figure 11, Figure 12 and Figure 13, we present, for every model, the distribution of FIDs and the corresponding samples.

Figure 10: MNIST: Distribution of FIDs and corresponding samples for each model when sampling parameters from *wide* ranges.

Figure 11: FASHION-MNIST: Distribution of FIDs and corresponding samples for each model when sampling parameters from *wide* ranges.

Figure 12: CIFAR10: Distribution of FIDs and corresponding samples for each model when sampling parameters from *wide* ranges.

Figure 13: CELEBA: Distribution of FIDs and corresponding samples for each model when sampling parameters from *wide* ranges.

# D Hyper-parameter Search over Narrow Ranges

In Figure 4 we presented the sensitivity of GANs to hyperparameters, assuming the samples are taken from the wide ranges (see Table 3). For completeness, in Figure 14 we present a similar comparison for the narrow ranges of hyperparameters (presented in Table 4).

Figure 14: A *narrow range* search of hyperparameters which were selected based on the wide hyperparameter search on the FASHION-MNIST data set. Black stars indicate the performance of suggested hyperparameter settings. For each model we allow 50 hyperparameter samples. From the practical point of view, there are significant differences between the models: in some cases the hyperparameter ranges *transfer* from one data set to the others (e.g. NS GAN), while others are more sensitive to this choice (e.g. WGAN). We note that better scores can be obtained by a wider hyperparameter search.

Figure 15: For each model we search for best hyperparameters on the *wide range*. Then, we retrain each model using the best parameters 50 times with random initialization of the weights, keeping everything else fixed. We observe a slight variance in the final FID. Hence, when an FID is reported it is paramount that one compares the entire distribution, instead of the *best* seed for the best run. The figure corresponds to Table 2.

# E  Precision, Recall and $F_1$ as a Function of the Budget

Figure 16: Optimizing for $F_1$, threshold $\delta = 1.0$.

Figure 17: Optimizing for $F_1$, threshold $\delta = 0.5$.

Figure 18: Optimizing for precision, threshold $\delta = 1.0$.

# F  Impact of the optimization algorithm.

We ran the WGAN training across 100 hyperparameter settings. In the first set of experiments we used the ADAM optimizer, and in the second the RMSPROP optimizer. We observe that distribution of the scores is similar and it's unclear which optimizer is "better". However, on both data sets ADAM outperformed RMSPROP on recommended parameters (CIFAR10: 154.5 vs 161.2, CELEBA: 97.9 vs 216.3) which highlights the need for a hyperparameter search. As a result, the conclusions of this work are not altered by this choice.

Figure 19: Optimizing for precision, threshold $\delta = 0.5$.

Figure 20: Optimizing for recall, threshold $\delta = 1.0$.

Figure 21: Optimizing for recall, threshold $\delta = 0.5$.

# G $F_1$, precision, and recall correlation with FID

Figure 22: Correlation of FID with precision, recall, and $F_1$. We observe that the proposed measure is particularly suitable for detecting a loss in recall.