[Reviews · NeurIPS 2018]

Reviewer 1



This paper introduces a large set of experiments to compare recently proposed GANs. It discusses two previously proposed measures -- inception score (IS) and Frechet Inception Distance (FID); and it proposes a new measure in the context of GAN assessment, based on precision, recall and F1. Precision (P) is measured as the fraction of generated samples with distance below a pre-defined threshold \delta; while recall (R) is measured as the fraction of inversely generated samples (from test set) with squared Euclidean distance below \delta (F1 is the usual mean between P and R). The paper argues that IS only measures precision and FIS measures both, so IS is essentially dropped as a measurement for GANs. Then the paper argues that it is important to show the mean and variance of FID and P-R-F1 measurements instead of the best values, computed over a set of random initialisations and hyper-parameter search points. This is an extremely important point that is often overlook by several papers published in the field, so I completely agree with the paper in this aspect. The paper also shows the performance of each GAN in terms of computational budget (not clear in the paper what this is, but I'm assuming some fixed amount of GPU time, for each tick in the graphs) and conclude that all models can reach similar performance with unlimited computational budget. The paper also reaches these conclusions: 1) FID cannot detect overfitting to the training data set, and 2) many dimensions have to be taken into account when comparing different GANs, but the paper only explores a subset of them. I think that it is laudable that a paper tries to address this challenging task, and the results are potentially useful for the community. I think that the experimental setup is interesting and compelling, but hard to be replicated because that would require a very large amount of computational resources that is not available for the majority of researchers. The main conclusion of the paper is expected (that there is really no model that is clearly better than others in all conditions and for all datasets), but not very helpful for the practitioner. There are interesting points, particularly considering the P-R-F1 measure -- for instance, looking at Fig. 5, it seems that NS GAN is a clear winner, but there is no strong comment about that in the paper, so I wonder if there is any flaw in my conclusion. Also, what is the correlation between F1 and FID measures? Other issues: - In Figure 1, Why does the text say "rather high bias", while the figure caption says "slight bias"? How is it possible to have a bias much bigger than the variance from samples of arguably the same set? - Please clarify what budget means in Figure 3. - Where is the VAE result in Figure 5?

Reviewer 2



In this paper the authors tackle the non-trivial problem of GANs comparison. Comparing the different approaches toward training GANs is a major issue towards a better understanding of deep generative models. This paper introduces a methodological procedure for comparing GANS via two different metrics FIA, and Precision, recall and F1. The authors compare 7 Different GANs (different Generative/Discriminator loss, and regularization terms) under the same network architecture. A procedure for testing different hyperparameters, that allows for different computational budgets, is introduced. The randomness due to network initialization is also analyzed. The experiments are run on four popular datasets, and also a synthetic dataset consisting of binary images of triangles is used. This synthetic dataset allows to directly compute the distance from a generated sample to the considered manifold (via non-convex optimization). There are two main aspects that show the importance/contributions of the paper. Introduces a methodology for comparing GANs (choose a bunch of different loss functions, different regularizations, fix architecture, adopt a metric, choose different datasets, repeat experiments since training involve randomness, introduce the computational budget as a way of quantitatively comparing how difficult is to reach good hyperparameters); Reaches interesting conclusions. As a first example, the paper concludes from empirical evidence that it does not make sense to compare the best FID results. Comparison should involve the distribution of the adopted metric under different runs. As a second example, the authors show empirically that given the right hyperparameters the best GAN (according to FID) depends on the dataset. There is no clear winner. I believe this paper will help the community to start a discussion on how to fairly compare deep generative models. *Major Points* The authors claim that algorithmic differences in state-of-the-art GANs become less relevant, as the computational budget increases. This is true according to the way the authors carried out the experiments. But, what happens if we start the hyperparameters tuning by doing a random search from the recommended values for each algorithm. Maybe the hyperparameters recommended by the authors are already good enough (or they are close to the best ones) for that particular dataset. For instance, according to the authors of WGAN-GP the same set of hyperparameters worked OK in many different experiments. Please comment. A clear limitation of the current work (in terms of the conclusions that are drawn) is that only one architecture is considered (infoGAN). All the conclusions are with respect to this architecture. So, one wonders whether this analysis is general enough. The implicance of this should be commented in the text. The conclusions are too strong. In fact, the paper shows that the average performance (across datasets) is similar but the best one depends on the dataset. This does not mean that all the GANs will reach the same best performance on every dataset. Inability to detect overfitting by FID. This is commented in Section 5, but then it is not linked to the rest of the paper. In particular, the finding regarding precision and recall on the synthetic dataset are not commented in the conclusions. I recommend to improve the connection of this experiment to the large-scale experimental evaluation. *Minor Comments / Typos* l26 - metric → metrics l40 - Fréchet Inception Distance → Fréchet Inception Distance (FID) l77 - mean and covariance is estimated → mean and covariance are estimated l95 - "... and corresponding loss, and the presence..." → "... and corresponding loss, and (ii) the presence..." l103 - MM GAN and WGAN should be in \textsc l119 - Add (\textsc{began}) somewhere in the sentence. l93-l120 - What about the different hyperparameters that only a few configurations have: $c$, $k_t$, $\gamma$. l155- In general when refer to a Figure/Table in Appendix please say that it is in the Appendix/Supplementary Material. l225- How bad is to get stuck in a local minima when doing this minimization? It seems that this implies that you are giving an upper bound on the performance (i.e., false negative rate might be lower). How many times did you repeat from different initializations. Please comment. l230- Which dimension did you consider for the latent space (d_z)? l239 - " as a popular alternative" → "as a popular alternative to GANs" l294- "with distance below" → "with distance to the manifold below" **Updated After Rebuttal** I see three main limitations of the paper: 1. Only one architecture is tested, 2. Hyperparameters search is not completely fair (in rebuttal the authors mention sequential Bayesian optimization as FW on this matter), 3. Very hard to replicate due to the large amount of computational resources needed. Even with 1. and 2., I think the paper worths publication since one can build on this work to compare different architectures and more rational ways of searching for hyperparameters. Regarding 3, the authors open sourced code and trained models which is a step forward on reproducibility. Finally the current conclusions of the paper are too strong (this point needs to be revised).

Reviewer 3



The paper aims to offer a consolidating view of the different GAN architectures and aims to establish comparative baseline to establish comparison of their performance using large sclae study using statistical measures. It also tries to establish the need to report distribution of FID results (as opposed to best) with fixed computation budget, owing to randomization and model instability. It goes onto discussion on the metrics and covers IS and FID and their pros/cons, and how FID is robust to mode dropping, and use of different enconding network and best FID achievable on classic datasets. It also contributes manifold of complex polygons as e.g's of one of the tasks (for which the manifold can be easily computed) that can be used for computing approximate precision and recall, F1 for comparing different GAN arechiectures that helps in surfacing overfitting (one of the weaknesses of both IS and FID). The authors also maintain they released their experimental setup as open source. Various DCGAN style architecture GANs are evaluated using above measures and setup (same archiecture, and corresponding loss function) on low-to-medium complexity datasets (MNIST,Fashion MNIST, CIFAR, CELEBA) to arrive at the conclusion that there is no statistical difference (based on FID, given enough computation budget) in the different GANs from the original (minimax and non-saturating style) GAN's proposed by Goodfellow et. al. The large variation in results owing to sensitivity of GAN's to hyper parameters and dataset is also highlighted. Main critiques - in terms of describing the GAN's the authors mention the make-up includes architecture, loss function and recipe/parameters, thus, when utilizing the loss function while keeping the archiecture same (for comparitive purposes), it seems rather than claiming that there were no statistically significant different across different GAN's for this 'neutral' setup, a more appropriate one would be for same choice of archiecture, the loss functions from different GAN's offer no statistically significant difference based on the setup and datasets evaluated - the choice of the parameters such as the learning rate is rather on higher side (any reason for that) -in some cases, such as Table 2, it appears that WGAN or WGAN GP has lowest FID across all datasets, I don't see mention of that in the results discussion on page 7